# Magnetic Metal Organic Framework Immobilized Laccase for Wastewater Decolorization

**Abdelfattah Amari** [1,2,*], **Fatimah Mohammed Alzahrani** [3,*], **Norah Salem Alsaiari** [3,*], **Khadijah Mohammedsaleh Katubi** [3,*], **Faouzi Ben Rebah** [4,5,*] and **Mohamed A. Tahoon** [4,6]

1 Department of Chemical Engineering, College of Engineering, King Khalid University, Abha 61411, Saudi Arabia
2 Research Laboratory of Energy and Environment, Department of Chemical Engineering, National School of Engineers, Gabes University, Gabes 6072, Tunisia
3 Chemistry Department, College of Science, Princess Nourah bint Abdulrahman University, Riyadh 11671, Saudi Arabia
4 Department of Chemistry, College of Science, King Khalid University, P.O. Box 9004, Abha 61413, Saudi Arabia; tahooon_87@yahoo.com
5 Higher Institute of Biotechnology of Sfax (ISBS), Sfax University, P.O. Box 263, Sfax 3000, Tunisia
6 Chemistry Department, Faculty of Science, Mansoura University, Mansoura 35516, Egypt
* Correspondence: abdelfattah.amari@enig.rnu.tn (A.A.); fmalzahrani@pnu.edu.sa (F.M.A.); nsalsaiari@pnu.edu.sa (N.S.A.); kmkatubi@pnu.edu.sa (K.M.K.); benrebahf@yahoo.fr (F.B.R.)

**Abstract:** The laccase enzyme was successfully immobilized over a magnetic amino-functionalized metal–organic framework $Fe_3O_4$-$NH_2$@MIL-101(Cr). Different techniques were used for the characterization of the synthesized materials. The $Fe_3O_4$-$NH_2$@MIL-101(Cr) laccase showed excellent resistance to high temperatures and low pH levels with a high immobilization capacity and large activity recovery, due to the combination of covalent binding and adsorption advantages. The long-term storage of immobilized laccase for 28 days indicated a retention of 88% of its initial activity, due to the high stability of the immobilized system. Furthermore, a residual activity of 49% was observed at 85 °C. The immobilized laccase was effectively used for the biodegradation of Reactive Black 5 (RB) and Alizarin Red S (AR) dyes in water. The factors affecting the RB and AR degradation using the immobilized laccase (dye concentration, temperature and pH) were investigated to determine the optimum treatment conditions. The optimum conditions for dye removal were a 5 mg/L dye concentration, temperature of 25 °C, and a pH of 4. At the optimum conditions, the biodegradation and sorption-synergistic mechanism of the $Fe_3O_4$-$NH_2$@MIL-101(Cr) laccase system caused the total removal of AR and 81% of the RB. Interestingly, the reusability study of this immobilized enzyme up to five cycles indicated the ability to reuse it several times for water treatment.

**Keywords:** wastewater treatment; metal–organic framework; laccase immobilization; adsorption; biodegradation

## 1. Introduction

Recently, the natural ecosystem has been greatly affected by the discharge of organic dyes, which are extensively used in the textile industry. Textile effluents directly discharged into natural water sources cause health problems [1]. Reactive Black 5 (RB) and Alizarin Red S (AR) dyes are two of the most harmful dyes used in textile manufacturing [2,3], utilized to give color to specific products. Various dyes contain in their structure phenolic rings and several functional groups, like $NO_2$, NO, OH, COOH, $NH_2$, NHR and $NR_2$, which are responsible for their toxicity [4,5]. Therefore, the resulting dye effluents could be toxic, carcinogenic or mutagenic to aquatic and human life [6,7]. Additionally, dyes can make water colored, blocking sunlight penetration at the water's surface, inhibiting photosynthesis and damaging the ecosystem.

Dye removal from wastewater can be achieved using various methods, such as photocatalysis, coagulation and adsorption [8–10]. However, these methods are limited due to the high cost of processing, lower removal capacity and the technical difficulties at a large scale. Recently, great attention has been paid to improving the efficiency of the biodegradation methods for removing dyes from water due to various advantages (e.g., high efficiency, low cost, facility and green processing). In this context, the use of enzymes for dye removal has attracted considerable attention and reached significant results. Among all enzymes, laccases have great biotechnology applicability due to their excellent catalytic activity and substrate specificity [11]. However, the industrial application of enzymes is hampered by their difficult recovery and their activity loss related to the ease of distortion of their 3D structure under inadequate operating conditions (solution type, temperature and pH) [12]. The biocatalytic activity of laccases can be improved by their immobilization using the suitable material [13]. Different methods, such as covalent linkage, adsorption and encapsulation are used to immobilize laccases on the proper material [14]. However, strong chemical interactions between the enzyme and the support with superior performance were achieved using the covalent linkage method [15]. For instance, graphene nanosheets have been applied for laccase immobilization with greater recycling performance via covalent bonding than physical adsorption [16]. In addition, glutaraldehyde has been used to link polymicrospheres (VFA-co-EGDMA) and laccase enzymes, allowing great improvement of enzymatic activity [17]. For the same purpose, covalent bonding was used to immobilize laccases over carbon nanotubes [15]. These studies demonstrate that laccase recycling efficiency, acid–base tolerance and thermal stability can be reached via the covalent linkage approach. Inorganic materials are the most widely used substrates for laccase immobilization. Due to their very low surface area, inorganic materials offer a very low loading capacity of laccase (10–100 $mg/g^{-1}$) [12]. As a result, their application for enzyme immobilization is limited due to the difficulty of controlling their pore size and surface distribution. In this context, the fabrication of proper material for the high loading affinity and high loading capacity of enzymes is a challenge. Metal–organic frameworks (MOFs) are crystalline porous materials including metal ions and organic ligands. MOF materials have tunable surface chemistries and pore sizes, as well as large specific surface areas [18–20], allowing their application for enzyme immobilization. For example, laccase immobilization was achieved using micro- and mesoporous Zr-MOF, achieving an improvement of the enzyme's reusability and stability [21]. MOF-immobilized enzymes showed hard recyclability and separation, which prevented their application in pollutant removal from water. These limitations can be overcome through the introduction of magnetism. Several magnetic MOFs were used for the removal of dyes [22] and drugs [23] from wastewater. Therefore, the combination between (1) a new class of materials (MOFs) that provides little diffusion limitation, excellent chemical and thermal stability, tunable porosity and a large surface area, (2) magnetic nanoparticles ($Fe_3O_4$) that provide magnetic separation, and (3) a laccase enzyme that has the ability to oxidize organic molecules and dyes can provide a promising material for water treatment. The covalent linkage between the MOFs and the biomolecules can be reached using functional groups as reported in the literature [24,25]. Laccase enzyme immobilization can be successfully reached via the introduction of functional groups to the MOFs [26,27]. However, many more studies are needed to demonstrate the applications of immobilized enzymes over magnetic functionalized MOFs for the removal of various pollutants from wastewater. Therefore, in this work, we synthesized magnetic amino-functionalized MOFs ($Fe_3O_4$- $NH_2$@MIL-101(Cr)) to immobilize the laccase enzyme for wastewater decolorization of both Reactive Black 5 and Alizarin Red S under different conditions, such as the dye concentration, pH and temperature. The immobilization parameters were determined, and the characters of the free and immobilized laccase were compared. Interestingly, the immobilized material was shown to be very promising for wastewater decolorization.

## 2. Materials and Methods

### 2.1. Chemicals

The 2,2-azinobis-3-ethylbenzothiazoline-6-sulfonate (ABTS), Reactive Black 5 (RB) Alizarin Red S (AR) and laccase from *Trametes versicolor* were supplied by Sigma-Aldrich. Anhydrous sodium acetate, ferric chloride hexahydrate ($FeCl_3 \cdot 6H_2O$) and ethylene glycol were supplied by Al. Nasr Co. (Cairo, Egypt). The 1,6-hexanediamine, terephthalic acid ($H_2BDC$) and chromium nitrate nonahydrate ($Cr(NO_3)_3 \cdot 9H_2O$) were supplied by Sinopharm Chemical Reagent Co., Ltd. (Shanghai, China). Distilled water was used for the preparation of all experimental solutions. All received chemicals were of analytical grade and used without any additional purifications.

### 2.2. Amino-Functionalized Magnetic Metal–Organic Framework Synthesis

The synthesis of the amino-functionalized magnetic metal–organic framework ($Fe_3O_4$-$NH_2$@MIL-101(Cr)) was performed according to the literature [28] with minor changes. Briefly, a mixture of 1,6-diaminohexane (10.5 mL), anhydrous sodium acetate (2.1 g), ethylene glycol (31 mL) and ferric chloride hexahydrate (1.1 g) was heated with continuous stirring for 11 h at 185 °C in a polytetrafluoroethylene liner to produce $Fe_3O_4$-$NH_2$. A total of 0.4 g of the obtained product ($Fe_3O_4$-$NH_2$) was washed, dried at 65 °C and dispersed in a mixture of terephthalic acid (1.68 g) and $Cr(NO_3)_3.9H_2O$ (4.2 g). Then, the mixture was heated at 215 °C (for 19 h) to produce the $Fe_3O_4$-$NH_2$@MIL-101(Cr). Finally, the resulting material was washed with $H_2O$ and dried at 105 °C.

### 2.3. Amino-Functionalized Magnetic Metal–Organic Framework Characterization

The synthesized $Fe_3O_4$-$NH_2$@MIL-101(Cr) was characterized using different instruments, including thermogravimetric analysis (TGA) measurement, a Fourier transform infrared (FT-IR) spectrophotometer and a scanning electron microscope (SEM). A Perkin Elmer STA 6000 (PerkinElmer Inc., Shelton, CT, USA) was used to measure the thermogravimetric analysis for the evaluation of thermal stability. A Bruker Tensor 27 FT-IR (Karlsruhe, Germany) spectrophotometer was used to measure the FT-IR spectra in the range of 500–4000 cm$^{-1}$. An SEM (Hitachi S4800, Hitachi, Tokyo, Japan) was used to study the morphology of the synthesized material.

### 2.4. Enzyme Immobilization

#### 2.4.1. Laccase Activity Assay and Immobilization Process

ABTS was used as a substrate to determine the laccase activity. A UVD-2960 UV spectrometer (Labomed Inc., Los Angeles, CA, USA) was used for measuring the ABTS oxidation product (ABTS$^+$) at 422 nm. Commonly, the reaction was performed between an appropriate volume of laccase and 1 mM ABTS for 5 min at 25 °C and a pH of 4 (adjusted using sodium acetate buffer). An absorbance increase was an indicator of laccase activity. The amount of laccase necessary to produce 1 mmol of ABTS$^+$ in a minute is known to be one unit of activity. The enzyme activity was studied at a pH range from 2 to 7 and at a temperature range from 25 to 85 °C.

For laccase immobilization over the synthesized substrate ($Fe_3O_4$-$NH_2$@MIL-101(Cr)), laccase liquid (designed amount) was suspended in an 11 mL acetate buffer (pH 4) with 3.5 mg of the support. Then, magnetic separation was used to obtain the immobilized material, which was shaken at 30 °C to reach the immobilization process. The immobilization yield and laccase loading were then determined by the collection of the supernatant.

#### 2.4.2. Optimal Immobilization Conditions

The influence of the immobilization time and enzyme dosage was determined to determine the optimum immobilization conditions. A time range from 1–8 h was used to study the influence of the immobilization time, while a laccase volume range from 0.5 to 3 mL was used to study the influence of the enzyme dosage with 3.5 g of the support

for 4 h at 30 °C in an 11 mL acetate buffer (pH 4). Then, the immobilization yield was determined using the following equation:

$$\text{Immobilization yield (\%)} = [(A_i - A_f)/A_i] \times 100 \tag{1}$$

where the initial and final laccase activity (U/mL) are denoted by $A_i$ and $A_f$, respectively.

### 2.4.3. Immobilized Laccase Stability

Immobilized and free laccases were stored for 28 days at 4.0 °C in a pH 4 solution (acetate buffer) to study the storage stability. Every four days, the laccase activities were determined. A 100% laccase activity was considered the initial activity. Immobilized and free laccases were kept for 6 h in the temperature range of 15–85 °C to determine their thermal stability. A 100% laccase activity was considered the initial activity at each temperature, and the final activity was also determined.

### 2.4.4. Kinetic Measurements

By using different concentrations of the substrate solution, and based on the oxidation reaction of ABTS, kinetic parameters ($V_{max}$ and $K_m$) that represented the maximum rate of reaction and Michaelis–Menten constant were measured. The immobilized and free laccases' apparent kinetic parameters were determined using a Lineweaver–Burk plot. The ABTS concentrations were 0.3–0.8 mmol/L. An acetate buffer (pH 4) was used to adjust the pH value, and the reaction temperature was 25 °C.

### 2.4.5. Dye Removal

RB and AR dye removal using $Fe_3O_4$-$NH_2$@MIL-101(Cr) immobilized laccase was investigated. For comparison, $Fe_3O_4$-$NH_2$@MIL-101(Cr) with inactivated laccase was investigated for the same purpose. Inactivation of the enzyme over support was achieved through incubation at a high temperature (80 °C) for 120 min, causing protein deactivation. Dyes with different concentrations (1, 5, 10 and 20 mg/L) with a volume of 10 mL were mixed with 50 mg of the immobilized active and inactive laccases in vials with continuous shaking for 24 h at 25 °C and a pH of 4 to determine the role of enzymes in the removal process. The effect of different parameters on the removal efficiency was determined. The dye removal at different temperatures of 5, 25, 35 and 50 °C was investigated at a pH of 4 with a dye concentration of 5 mg/L. The effect of the pH on dye degradation was investigated at pH values of 4, 5, 6 and 7 at 25 °C and a dye concentration of 5 mg/L. Additionally, the effect of the dye concentration on the removal efficiency was investigated using concentrations of 1, 5, 10 and 20 mg/L at a pH of 4 and temperature of 25 °C. After each experiment mentioned above, the magnetic $Fe_3O_4$-$NH_2$@MIL-101(Cr) immobilized laccase was separated using a magnet, and the solution was analyzed for the presence of dye. Furthermore, the reusability of the $Fe_3O_4$-$NH_2$@MIL-101(Cr) immobilized laccase was determined up to 5 successive cycles for the degradation of the dyes at pH 4, 25 °C, a dye concentration of 5 mg/L and a contact time of 24 h. After each studied cycle, the immobilized enzyme was separated using a magnet and washed several times with a buffer solution (pH 4) to be used in the next cycle. All experiments were performed in triplicate.

## 3. Results and Discussions

### 3.1. Nanocomposite Characterization

The magnetic metal–organic framework $Fe_3O_4$-$NH_2$@MIL-101(Cr) and immobilized laccase particle size and surface morphology were characterized using SEM analysis as shown in Figure 1a,b, respectively. The surface morphology of the metal–organic framework was not affected by laccase immobilization. The size of the particles remained the same after laccase immobilization in the range of 30–40 nm. According to the SEM images, the $Fe_3O_4$-$NH_2$@MIL-101(Cr) had a regular spherical shape before and after immobilization. Additionally, the presence of functional groups was determined using the FTIR spectra of the free laccase, immobilized laccase and $Fe_3O_4$-$NH_2$@MIL-101(Cr) MOF as shown in

Figure 1c. The $Fe_3O_4$-$NH_2$@MIL-101(Cr) MOF typical peaks are represented by sharp peaks at 590, 1400 and 1554 cm$^{-1}$ [28]. The amine stretching vibration of the C-N bond is represented by the peak band at 1321 cm$^{-1}$ [29]. For the laccase, $CH_2$ group stretching vibrations in the laccase protein are represented by the peak at 2921 cm$^{-1}$, while the N-H and OH bonds are represented by the band at 3681–3000 cm$^{-1}$ [30]. The laccase showed a number of other distinct protein peaks [31,32], like the bands at 1166–949 cm$^{-1}$, 800–500, 1421–1211 cm$^{-1}$ and 1643 cm$^{-1}$, corresponding to the protein in the laccase, out-of-plane NH and C-O bending of amide V and VI, CN stretching and NH bending of amide III, and C-O stretching of amide I, respectively.

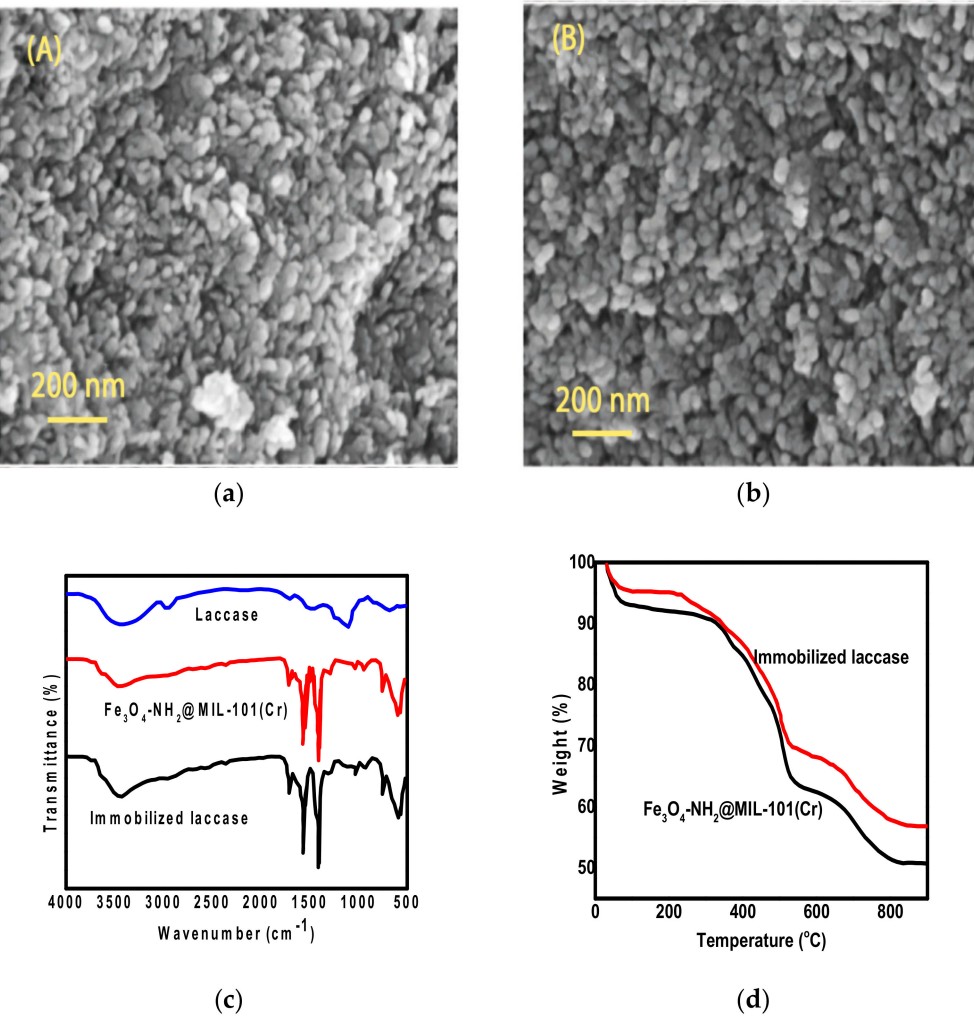

**Figure 1.** (**a**) SEM image of $Fe_3O_4$-$NH_2$@MIL-101(Cr). (**b**) SEM image of the immobilized laccase. (**c**) FT-IR of different materials. (**d**) TGA curves of $Fe_3O_4$-$NH_2$@MIL-101(Cr) and the immobilized laccase.

Laccase immobilization on $Fe_3O_4$-$NH_2$@MIL-101(Cr) is demonstrated by the wide peaks at 3400 cm$^{-1}$ showing the hydroxyl and amino group interaction. $Fe_3O_4$-$NH_2$@MIL-101(Cr) MOF was covalently bonded to the laccase enzyme, which was confirmed by the stretching vibrations of the C-O and C-N bonds appearing after immobilization at 1631 and 1421 cm$^{-1}$, respectively. The thermogravimetric curve of $Fe_3O_4$-$NH_2$@MIL-101(Cr) MOF and the immobilized laccase are shown in Figure 1d. According to Figure 1d, 211 °C represents the temperature at which weight loss started for the immobilized laccase, unlike the curve of $Fe_3O_4$-$NH_2$@MIL-101(Cr) MOF, which was also represented in another study [21]. This weight loss at 211 °C was an indication of laccase immobilization over the $Fe_3O_4$-$NH_2$@MIL-101(Cr) MOF, and this temperature represents the destruction of the interaction between them. A continuous weight decrease was observed as the

temperature increased due to the collapse of the $Fe_3O_4$-$NH_2$@MIL-101(Cr) MOF. Both curves of the MOFs and the immobilized laccase had the same behavior. The immobilized laccase thermogravimetric curve hit a plateau at about 805 °C, similar to that of the MOFs. The immobilized laccase kept 57% of its initial weight when the temperatures reached 900 °C, which was a little higher than that of the MOFs due to the laccase's remaining organic residues after its pyrolysis [29].

### 3.2. Optimal Immobilization Conditions

The time effect on laccase immobilization over the $Fe_3O_4$-$NH_2$@MIL-101(Cr) MOF surface was determined as shown in Figure 2a. During the first 3 h, quick immobilization occurred with a loading amount of 45.23 mg/g, and equilibrium was reached at 4 h with a loading of 69.19 mg/g. After 4 h, the immobilization yield stopped rising and ranged from 81.3% to 85.2%. The thermal effect of the immobilization reaction between the enzyme and the support with the crowding of laccase over the $Fe_3O_4$-$NH_2$@MIL-101(Cr) surface led to a drop in enzyme activity, and the immobilization yield varied between 4 and 8 h. Similar behavior was observed in the literature [33].

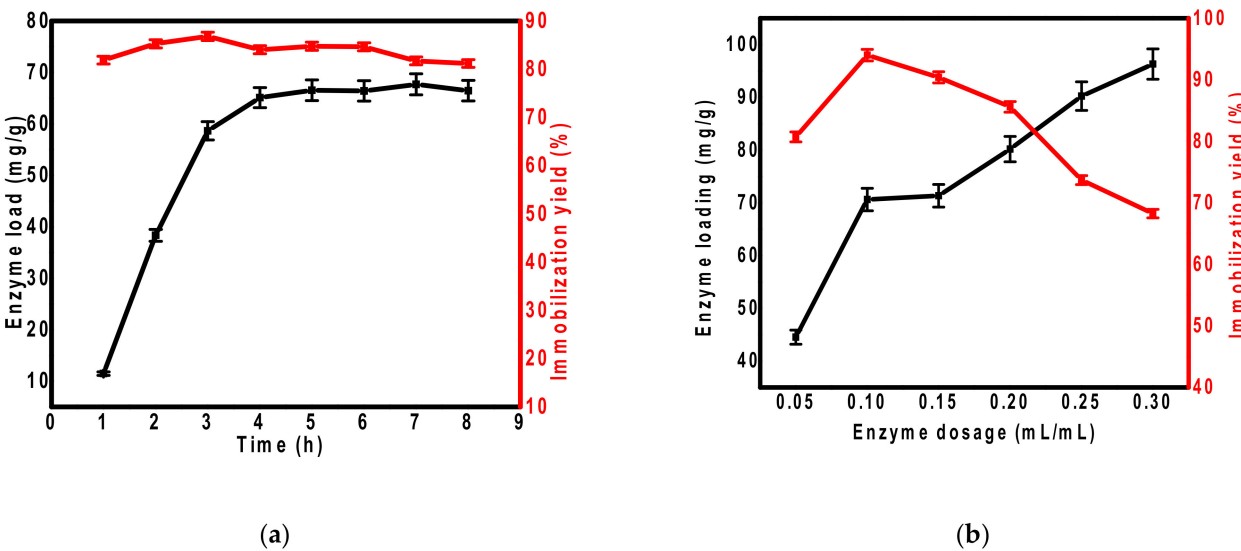

(**a**)                                                                                                    (**b**)

**Figure 2.** The effect of the (**a**) immobilization time and (**b**) enzyme dosage on enzyme loading and immobilization yield.

The dosage's effect on the laccase immobilization was also determined as shown in Figure 2b. According to Figure 2b, the laccase loading reached 70.5 mg/g, and the immobilization yield reached 93.9% when the enzyme dosage was 0.1 mL/mL. This behavior was attributed to the covalent bonds between the laccase and little amino groups of MOFs, as reported in a previous study [34]. After that, no additional enzymatic dose could be linked to the support that no longer contained any vacant sites. Hence, the immobilization yield decreased due to the decrease in immobilization efficiency. Consequently, the dose of 0.1 mL/mL was the optimum laccase amount. Briefly, the optimum time was 4 h, and the optimum dose was 0.1 mL/mL for the immobilization of laccase over the $Fe_3O_4$-$NH_2$@MIL-101(Cr) MOF. These excellent immobilization results were attributed to the MOFs being well-dispersed in water with no agglomeration. Additionally, the pore sizes of the MOFs help the good adsorption of laccase enzyme over their surfaces.

### 3.3. Immobilized Laccase Properties

#### 3.3.1. Thermal and Storage Stability

The immobilized and free laccase thermal stabilities were investigated and are shown in Figure 3a. According to Figure 3a, the laccase activity was greatly decreased with increasing temperatures. The free laccase relative activity was lower than that of the immobilized laccase. The free laccase retained 34% of its initial activity at 45 °C and

6 h of incubation, but the immobilized laccase retained 87%. At 85 °C, the free laccase was approximately deactivated while the immobilized laccase retained about 49% of its activity. Thus, the immobilization process improved the laccase's ability to resist elevated temperatures. The $Fe_3O_4$-$NH_2$@MIL-101(Cr) MOF and laccase interaction induced the laccase's protection from damage when exposed to the environment [35]. One of the most important factors for determining the ability to use an immobilized enzyme in different applications is its long-term storage stability. The long-term storage stability of the free and immobilized laccases is shown in Figure 3b.

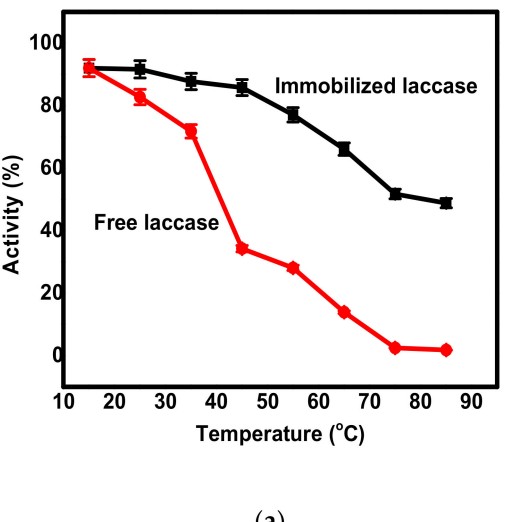

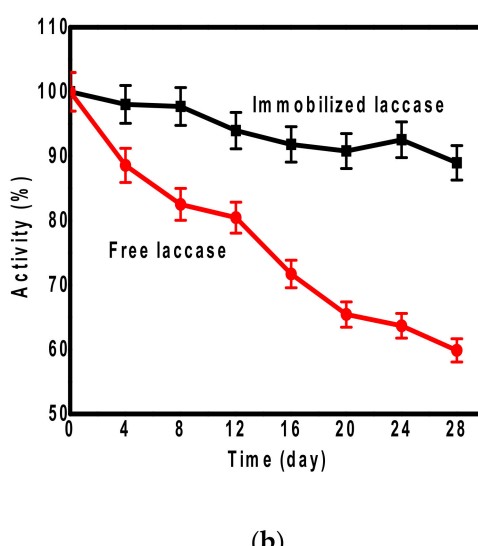

(**a**)           (**b**)

**Figure 3.** The effect of the (**a**) thermal stability and (**b**) storage time on the free and immobilized laccases.

According to Figure 3b, the free laccase retained 80% of its activity after a 10-day storage period, while the immobilized laccase retained 95% of it activity within the same time. At the end of storage time (28 days), the free laccase retained 61% of its initial activity, while the immobilized one retained 88% of its activity. Noticeably, the rate and value of activity loss for the free laccase was higher than the immobilized laccase. The laccase was less prone to deactivation due to the moderately stable environment provided by immobilization. The storage stability of the immobilized laccase over the $Fe_3O_4$-$NH_2$@MIL-101(Cr) was higher than that in a previous report [36]. The MOF and laccase interaction was achieved via covalent bonding and physical adsorption that protected the laccase's secondary and tertiary structure during long-term storage from damage, while the physical adsorption not only provided the same level of protection showed in the immobilized laccase over Zr-MOF [21]. Thus, the immobilization strategy was approved for long-term storage. Generally, the immobilized laccase showed higher tolerance than the free enzyme toward elevated temperatures and long-term storage.

### 3.3.2. Temperature and pH Effect

In all cases, the free enzyme had lower enzymatic activity than that of the immobilized enzyme. The temperature effect on the free and immobilized laccases was investigated as shown in Figure 4a. According to Figure 4a, the free laccase showed a lower temperature activity profile than the immobilized laccase. The free laccase showed an optimum temperature of 55 °C, while the immobilized laccase's was at 65 °C. This high optimum temperature shift may be related to the interaction between MOFs and the laccase enzyme. The interaction raised the activation energy, allowing the system to identify the best conformation for substrate binding. Additionally, at a temperature range of 55–85 °C, the immobilized enzyme retained its activity over 76%. Additionally, at 85 °C, the free laccase activity was approximately deactivated, while the immobilized laccase retained above 81% of its activity. These results show that the immobilized laccase had improved tolerance to

elevated temperatures compared with the free laccase. This could be attributed to the little variation at high temperatures of the laccase conformation due to the improved rigidity of the laccase structure by immobilization [37].

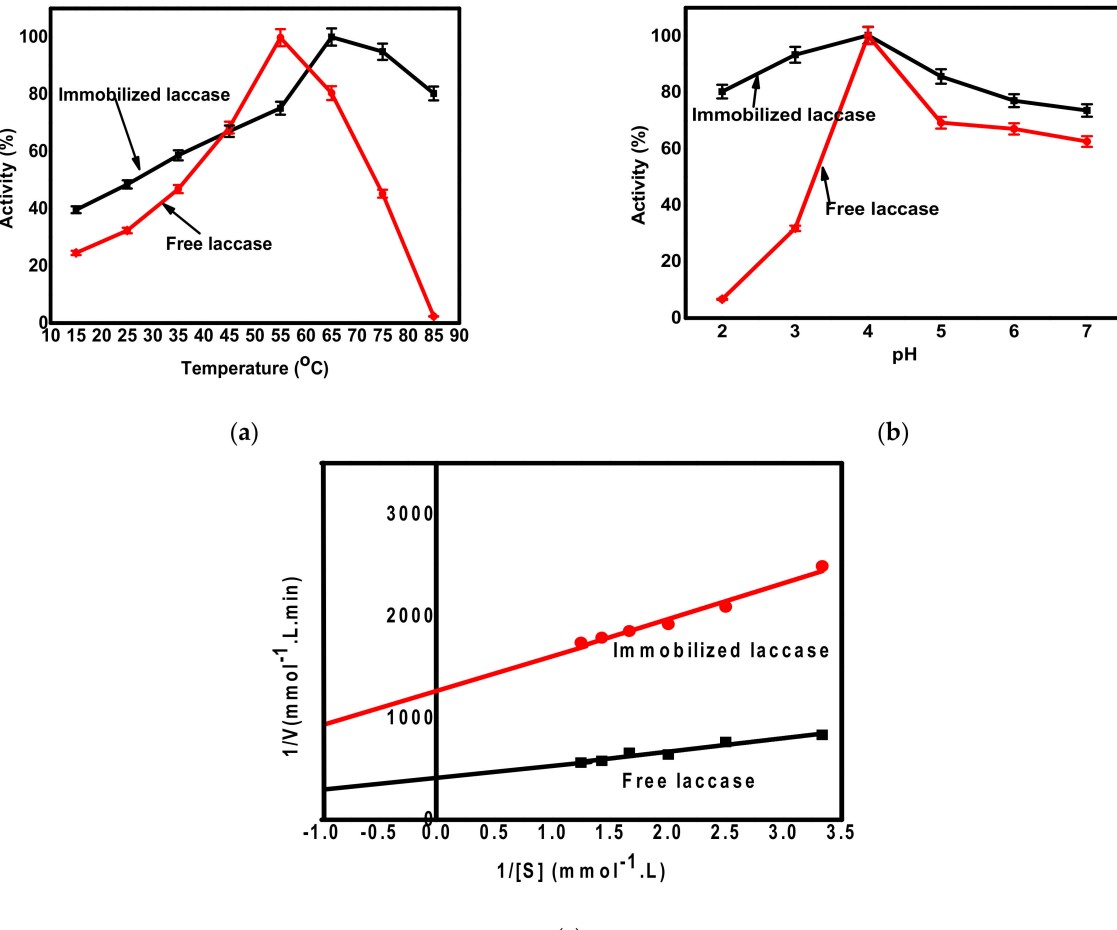

**Figure 4.** (**a**) The effect of the temperature and (**b**) pH and (**c**) the Lineweaver–Burk plots of the free and immobilized laccases.

The pH's effect on the enzymatic activity of the free and immobilized laccase was also investigated, as shown in Figure 4b. According to Figure 4b, the immobilized and free laccases had optimum activity at a pH of four. The free laccase showed a lower pH profile than the immobilized laccase. At a pH of two, the free laccase showed an activity of 8%, while the immobilized laccase showed an activity of 81% at the same pH. This stability of the immobilized laccase resulted from the several points of interaction between the amino groups of the MOFs and the laccase molecules [38]. In addition, a favorable microenvironment for catalysis was provided by the negatively charged amino groups on the MOFs. These amino groups may act as a buffer, allowing the pH to be higher in the microenvironment of the enzyme and become adequate for its activity [39]. Consequently, this immobilized laccase over the $Fe_3O_4$-$NH_2$@MIL-101(Cr) MOFs is suitable for water treatment applicability due to its resistance to high temperatures and low pH.

### 3.3.3. Kinetic Parameters

The immobilized and free laccase kinetic parameters were determined by calculating $K_m$ and $V_{max}$ as shown in Figure 4c. The $K_m$ values for the immobilized and free laccase were 0.7 and 0.1 mM, respectively, indicating the multiplication of the value by immobilization. This fact occurred due to the decreased laccase substrate affinity, as reported in the literature [40]. This multiplication of the $K_m$ value due to the restricted freedom of the

enzyme molecule resulted from the stability of the laccase enzyme over the MOFs' surface. After immobilization, the substrate diffusion restriction and the space steric hindrance of the laccase were created, resulting in a substantial decrease in the affinity between the ABTS and the enzyme. Furthermore, immobilization-induced structural changes in the laccase can lead to an increase of the $K_m$ value [41]. The $V_{max}$ was also reduced to 1.2 from 1.9 μmol/(L/min) due to the same previous factors, indicating that the immobilization retained 65% of the laccase–substrate reaction's maximum rate.

### 3.4. Dye Removal

3.4.1. AR and RB Dye Removal Using Inactivated Laccase

The AR and RB dyes, with concentrations of 1, 5, 10 and 20 mg/L, were used to study dye removal using thermally inactivated laccase as shown Figure 5a. According to the obtained results, the increase of the dye concentrations allowed a decrease of the sorption efficiency. Here, the dye removal resulted from the adsorption capacities of the $Fe_3O_4$-$NH_2$@MIL-101(Cr) MOFs toward the AR and RB dyes. The existence of functional groups of MOF surfaces is responsible for AR and RB dye capturing. Additionally, the presence of MOF pores helped in the chelation process of the contaminant dyes. Herein, the increased dye concentrations caused the saturation of adsorbents by the dye molecules, leading to the decrease in sorption efficiency. Interestingly, the results indicated the ability to use $Fe_3O_4$-$NH_2$@MIL-101(Cr) MOFs in water treatment.

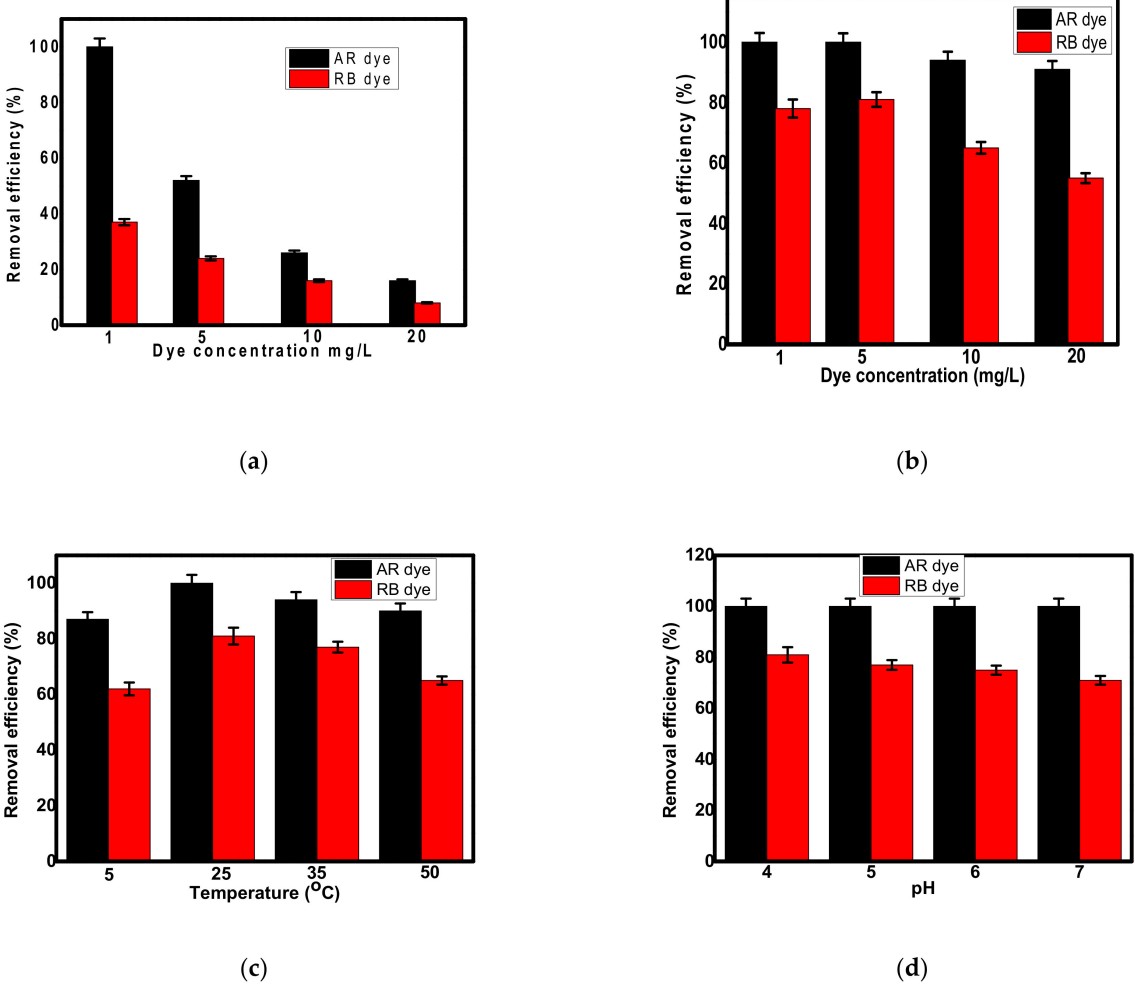

**Figure 5.** (**a**) Dye removal using inactivated laccase. (**b**) Dye concentration effect, (**c**) temperature effect and (**d**) pH effect on AR and RB removal over the $Fe_3O_4$-$NH_2$@MIL-101(Cr) laccase.

### 3.4.2. Dye Concentration Effect on Removal Efficiencies

The effect of different parameters, including the AR and RB concentrations, temperatures, and pH, on the removal capacity of the $Fe_3O_4$-$NH_2$@MIL-101(Cr) immobilized laccase was determined after investigating the sorption capacity via the inactivated laccase. The different concentrations of AR and RB dyes, ranging from 1 to 20 mg/L, were used to study the effect of the dye concentration on biodegradation, using laccase immobilized on $Fe_3O_4$-$NH_2$@MIL-101(Cr) as shown in Figure 5b. According to Figure 5b, the dye degradation rates decreased with increasing dye concentrations. The obtained results indicated that the laccase enzyme was well immobilized over the proper support $Fe_3O_4$-$NH_2$@MIL-101(Cr), allowing the total removal of AR and 81% of the RB from the water at an initial concentration of 1 mg/L. However, for the initial dye concentration of 20 mg/L, the removal efficiency reached 91% and 55% of the AR and RB, respectively. When comparing the results of the inactivated laccase (Figure 5a) and $Fe_3O_4$-$NH_2$@MIL-101(Cr) laccase (Figure 5b), it is clear that the second system provided higher removal efficiencies toward AR and RB dyes, indicating that the enzyme assisted in the degradation process. The removal efficiencies of the AR dye were 100%, 100%, 94%, and 91% at dye concentrations of 1, 5, 10 and 20 mg/L, respectively, while they were 78%, 81%, 65% and 55% for the RB dye at the same range of concentrations, respectively. The decreased efficiency of the $Fe_3O_4$-$NH_2$@MIL-101(Cr)-laccase system as a result of increasing the dye concentration, especially in case of the RB dye, was attributed to the inadequate amount of laccase over the MOF surface, as well as the dyes' complicated structures opposite these inadequate amounts of laccase that inhibit the total removal of these contaminants at higher concentrations [42].

### 3.4.3. Temperature Effect on Removal Efficiencies

A temperature range from 5 to 50 °C was used to study the temperature's effect on AR and RB removal, using immobilized laccase over $Fe_3O_4$-$NH_2$@MIL-101(Cr) as shown in Figure 5c. According to Figure 5c, 25 °C allowed the highest removal efficiencies of both dyes, being 100% for AR and 81% for RB. Compared with the removal efficiencies at 25 °C, at 5 °C, the removal values reached 87% and 62% for the AR and RB dyes, respectively. However, the temperature increasing to 50 °C led to the efficiency decreasing by 9% and 15% for the AR and RB, respectively. The high temperature activity of the laccase is attributed to the inhibition of laccase's thermal deactivation due to the stabilized enzyme structure by the $Fe_3O_4$-$NH_2$@MIL-101(Cr) support. While at a low temperature, the dye removal efficiencies were decreased due to the laccase's conformational changes, which happened at low temperatures [43], decreasing its activity. Nonetheless, even at high temperatures, degradation of the AR and RB dyes occurred more effectively than mere sorption, indicating that the enzyme was active and operated even at 50 and 5 °C. In a similar study, the laccase was immobilized over different supports like polyacrylamide gel, alginate-gelatin mixed gel and Ca-alginate for the removal of Aniline Blue, Trypan Blue, Congo Red and Methyl Red [44]. In this study, the effect of the temperature was determined in the range of 30–60 °C, and likely, the highest efficiency was achieved at 50 °C, indicating that the immobilization process maintained the enzyme structure and stability and increased its applicability for industrial water treatment. While the $Fe_3O_4$-$NH_2$@MIL-101(Cr) had strong sorption properties, the immobilized laccases had the higher dye removal performance. Thus, dye degradation was achieved through a dual sorption/enzymatic mechanism, as reported in the literature [45], for the removal of Acid Orange 7 using immobilized laccase on different lignocellulosic wastes.

### 3.4.4. The Effect of pH on Removal Efficiencies

The pH's effect on the AR and RB dye removal from water using $Fe_3O_4$-$NH_2$@MIL-101(Cr) immobilized laccase was studied at a pH range of 4–7 as shown in Figure 5d. According to Figure 5d, the dye removal was not greatly affected by the pH. At all pH values, the AR dye was removed completely from the water using the immobilized laccase, while RB dye removal decreased slightly with the pH. The highest dye removal

was achieved at a pH of four. According to the results, the dyes were degraded using $Fe_3O_4$-$NH_2$@MIL-101(Cr) immobilized laccase at different pH values, indicating the great protection of laccase by the MOF against environmental pH changes [46].

### 3.4.5. $Fe_3O_4$-$NH_2$@MIL-101(Cr) Laccase Reusability

Reusability is an important economic factor for determining the cost of the water treatment process [47,48]. In addition, reusability determines the applicability of the materials in industrial water treatment. Herein, it is important to study the reusability of the biocatalytic system's $Fe_3O_4$-$NH_2$@MIL-101(Cr) immobilized laccase for the removal of AR and RB dyes from water (Figure 6). The reusability study of $Fe_3O_4$-$NH_2$@MIL-101(Cr) immobilized laccase was performed in up to five operating successive cycles. According to Figure 6, the removal efficiencies in the first cycle for both dyes were the highest, due to the fresh, unused $Fe_3O_4$-$NH_2$@MIL-101(Cr) immobilized laccase with high degradation and adsorption capacities.

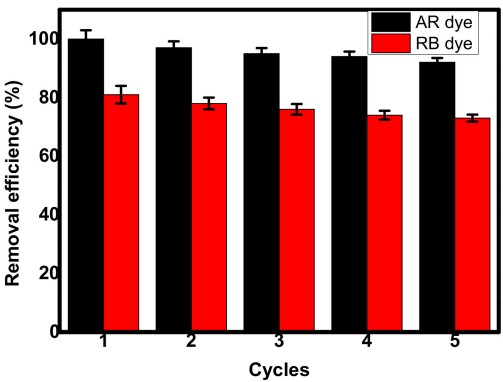

**Figure 6.** Reusability of $Fe_3O_4$-$NH_2$@MIL-101(Cr) laccase for AR and RB dye removal for up to five cycles.

It is very clear that the dye degradation efficiencies decreased from the first to the third cycle, and after that, the degradation rate became approximately constant. After the fifth cycle, the degradation efficiencies were 92% (for AR) and 73% (for RB), indicating a drop in the efficiencies by 8% and 12%, respectively. This fact approved the ability to reuse this immobilized laccase system for water treatment. The decreased efficiencies of the $Fe_3O_4$-$NH_2$@MIL-101(Cr) laccase toward the degradation of the AR and RB dyes were attributed to the activity loss of the laccase enzyme after each cycle. The laccase activity loss was attributed to the substrates, the degradation products and the changes in temperature and pH during the reaction. Additionally, the material washing after each cycle led to the loss of large amounts of laccase enzymes from the surface of the $Fe_3O_4$-$NH_2$@MIL-101(Cr) MOFs, which caused the decreased rate of dye degradation. Compared with previous studies, chitosan cross-linked with genipin was used to immobilize the laccase and applied for the degradation of dyes, with the retention of 48% of its efficiency after fourteen cycles [49]. Similarly, alginate-gelatin was used to immobilize the laccase for the degradation of dyes, with the retention of 65% of its activity after seven successive cycles [50]. This means that the $Fe_3O_4$-$NH_2$@MIL-101(Cr) laccase can be considered a promising process for dye removal form wastewater. For a full image of the $Fe_3O_4$-$NH_2$@MIL-101(Cr) laccase system, a future study about this immobilized material must be the determination of the thermodynamic [51–54] parameters of the degradation reaction.

### 4. Conclusions

In this study, laccase immobilization was achieved over $Fe_3O_4$-$NH_2$@MIL-101(Cr) as a magnetic amino-functionalized metal–organic framework. Different techniques, including SEM, TGA and FT-IR, were used for the characterization of the immobilized enzyme. The immobilization capacity reached 69.19 mg/g, due to the combination between the advan-

tages of covalent bonding and adsorption. The enzymatic properties of the immobilized laccase were enhanced (improved storage stability, high thermal stability and resistance to low pH levels). The kinetic parameters indicated the excellent affinity between the laccase and $Fe_3O_4$-$NH_2$@MIL-101(Cr). Additionally, the immobilized laccase was successfully used for the degradation of AR and RB dyes with high efficiency. During the dye degradation study, different operating factors were investigated, including dye concentrations, temperature and pH effects. Additionally, the dye removal using thermally inactivated laccase was investigated to determine the dye sorption parallel to the dye degradation. The results indicated the process's ability to completely remove the AR dye and 81% of the RB dye via sorption and biodegradation dual removal mechanisms. Moreover, the reusability of the immobilized laccase system for the removal of target dyes were investigated up to five cycles, showing slight efficiency decreases, with 92% and 73% removal rates for AR and RB, respectively, obtained after the fifth cycle. Consequently, this immobilized laccase can be considered a promising material for water treatment.

**Author Contributions:** A.A., N.S.A., F.M.A., K.M.K., F.B.R. and M.A.T.; methodology, A.A., N.S.A., F.M.A., K.M.K., F.B.R. and M.A.T.; formal analysis; data curation, A.A., N.S.A., F.M.A., K.M.K., F.B.R. and M.A.T.; writing–original draft preparation, A.A., N.S.A., F.M.A., K.M.K., F.B.R. and M.A.T.; writing—review and editing, A.A., N.S.A., F.M.A., K.M.K., F.B.R. and M.A.T.; visualization; F.B.R. All authors have read and agreed to the published version of the manuscript.

**Funding:** This research was funded by the Deanship of Scientific Research at King Khalid University.

**Acknowledgments:** The authors extend their appreciation to the Deanship of Scientific Research at King Khalid University for funding this work through the research groups program under grant number RGP.2/157/42. Additionally, this research was funded by the Deanship of Scientific Research at Princess Nourah Bint Abdulrahman University through the Fast-track Research Funding Program.

**Conflicts of Interest:** The authors declare no conflict of interest.

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
