# Peer review of "Magnetic Metal Organic Framework Immobilized Laccase for Wastewater Decolorization"

_processes, doi:10.3390/pr9050774_

Round 1

Author Response

  1. The introduction is poorly written and reads as though it has been translated from another language inaccurately, and it is a tedious read. Phrases like “ecosystem environment,” “enough efficient,” etc., should be removed. The authors must rewrite the entire introduction using the correct language so that it is readable.

The introduction was revised and modifications were highlighted in red.

  1. The paper is quite similar to Chemosphere, 233, 2019, 327-335, where they have studied the removal of phenolic compounds using the same laccase on magnetic/amino-functionalized MOFs. The authors do not cite that paper. There are numerous similar reports in the literature, which the authors do cite. I can not recommend this paper for publication in Processes as it seems to be an incremental work where they merely change the substrate. The authors will do well to rewrite the entire introduction before submitting elsewhere.

Is very important to indicate here that more investigations are needed to confirm what was reported in the literature. Also, much more studies are needed to demonstrate the applications of immobilized laccase over magnetic functionalized MOFs for the removal of various pollutants forom wastewater

Two references were added in the manuscript:

Wu E., Li Y., Huang Q., Yang Z., Wei A., Qi Hu Q.Laccase immobilization on amino-functionalized magnetic metal organic framework for phenolic compound removal. Chemosphere 2019, 233, 327-335

Qiu X.;Wang, Y.; Xue, Y.; Li, W.; Hu, Y. Laccase immobilized on magnetic nanoparticles modified by amino-functionalized ionic liquid via dialdehyde starch for phenolic compounds biodegradation Chemical Engineering Journal 2020, 391, 123564

Reviewer 2 Report

The authors have investigated the potential of magnetic metal-organic framework immobilized laccase for biodegradation of dyes in the water. In the introduction, the authors mentioned the negative effects of organic pollutants in the water on aquatic life and human health. The authors have well-argued the use of enzymes and metal-organic frameworks for water decolorization. The research presented in the manuscript has good quality. The Materials and Methods and Results and Discussion sections were written correctly and all necessary information has been provided. However, there are some language and style mistakes:
-the adverb “so” is often used at the beginning of the sentence. It should be changed to a more formal one.
-page 3, lines 114-117: please check the sentence. It is unclear.
-page 3, section 2.3: please write the equipment names in the same style (line 124).
-section 2.4: please unify the range of units. There is pH 4, 4.0, and 4.00 and time 4.0h.
-page 5: figure 1d, there should be a temperature unit.
-page 6, line 222: temperature unit.
-page 7: figure 3a, there should be a temperature unit.
-page 9, line 300-301: the sentence is unclear.
-page 10, line 338-339: the sentence is unclear.
The manuscript may be accepted for publication after a minor revision.

Author Response

-the adverb “so” is often used at the beginning of the sentence. It should be changed to a more formal one.

The adverb “so” was changed in the manuscript.

-page 3, lines 114-117: please check the sentence. It is unclear.

The sentence was re-written (Lines 118-121)

"0.4 g of the obtained product (Fe3O4-NH2) was washed, dried at 65 oC and dispersed in a mixture of terephthalic acid (1.68 g) and Cr(NO3)3.9H2O (4.2 g). Then, the mixture was heated at 215 oC (for 19 hours) to produce the Fe3O4-NH2@MIL-101(Cr). Finally, the resulted material was washed with H2O and dried at 105 oC."

-page 3, section 2.3: please write the equipment names in the same style (line 124).

The equipment names were written as required (section 2.3)

-section 2.4: please unify the range of units. There is pH 4, 4.0, and 4.00 and time 4.0h.

All the units were uniformed

-page 5: figure 1d, there should be a temperature unit.

The unit was added

-page 6, line 222: temperature unit.

The unit was added

-page 7: figure 3a, there should be a temperature unit.

The unit was added

-page 9, line 300-301: the sentence is unclear.

We have revised it.

-page 10, line 338-339: the sentence is unclear.

We have revised it.

Reviewer 3 Report

  • Some photos of reactor/sample are good for Material and Methods, especially for the result presentation
  • Chemical oxygen demand (COD) or Total organic Carbon (TOC) are addition if possible for good for showing the advanced oxidation process ability

Author Response

Some photos of reactor/sample are good for Material and Methods, especially for the result presentation

Chemical oxygen demand (COD) or Total organic Carbon (TOC) are addition if possible for good for showing the advanced oxidation process ability.

The experiments were conducted using samples of 10 mL volume in Erlenmeyer flask. Therefore, there is no photo of reactor that can be presented. Also, the methodology is well presented in the manuscript.

Now, we are planning experiment using larger volumes and with real wastewater from textile industries. In this context, various parameters will be evaluated such as the COD and the TOC.

Round 2

Reviewer 1 Report

The authors have addressed most of my comments and the paper can now appear in Processes